# Allocating constraint tasks schedules to promote on-demand cloud services by utilizing the Hungarian Algorithm

Nesreen Alsharman[1], Ismail Hababeh[1], Mohammad Alqudah[2], Kholoud Nairokh[1] and Deefallah Alshorman[3]

[1] Computer Science Department, German Jordanian University, Amman, Jordan
[2] Basic Sciences Department, German Jordanian University, Amman, Jordan
[3] Department of Elementary Teacher Education, Al-Zaytoonah University of Jordan, Amman, Jordan



Corresponding authors
Nesreen Alsharman,
nesreen.alsharman@gju.edu.jo
Ismail Hababeh,
ismail.hababeh@gju.edu.jo

## ABSTRACT

Cloud computing offers numerous benefits to its users, but it also presents significant performance challenges. The nondeterministic polynomial time (NP)-complete nature of cloud workflow scheduling makes it a significantly challenging task. Scheduling cloud tasks become considerably more complex when operations involve varying quality-of-service (QoS) requirements. Constrained workflow scheduling, however, has the potential to boost cloud system performance and consequently improve quality of service. Although numerous approaches have been developed for workflow scheduling, most focus exclusively on single QoS constraints. This study presents a method for utilizing the Hungarian Algorithm (HA) to address multiple workflow scheduling constraints and promote on-demand cloud services. The Fittest Task Population algorithm (FTPA) was developed to generate the fittest task population set that matches the customer tasks' constraints. The HA is utilized to assign each task in the generated fittest task population set to the fittest cloud virtual machine (VM). The proposed approach is validated and compared with state-of-the-art workflow scheduling methods using different multiple constraint scheduling scenarios. The comparative analysis validates the effectiveness of the proposed integrated FTPA-HA algorithm, demonstrating its superiority over existing scheduling approaches.

## INTRODUCTION AND BACKGROUND

Cloud computing has become a cornerstone of modern computing infrastructure, offering users dynamically scalable, flexible, and cost-efficient resources through Infrastructure-as-a-Service (IaaS) and related models (*Burak & Bharadiya, 2023*; *Noman, Fernand & Peter, 2022*; *Patel & Kansara, 2021*). The ability to provision resources on demand has attracted a wide range of applications, from large-scale data analytics and business intelligence to scientific workflows and real-time interactive services (*Albtoush et al., 2023*). Despite these benefits, scheduling workflow tasks in cloud environments remains a critical research

challenge. This challenge arises from the NP-complete nature of workflow scheduling problems, the heterogeneity of virtual machines (VMs), and the diversity of user-defined quality-of-service (QoS) constraints such as deadlines, cost, reliability, and energy efficiency. Addressing these challenges is essential for ensuring that cloud services meet user expectations while also maintaining provider efficiency and sustainability.

A considerable body of research has explored workflow scheduling in cloud computing, producing a wide range of strategies. These approaches can be broadly grouped into heuristic-based, metaheuristic-based, machine learning–based, and hybrid or multi-objective frameworks. Heuristic approaches rely on deterministic or rule-based techniques to allocate tasks to resources (*Hu, Wu & Dong, 2023*). Examples include list scheduling, earliest-finish-time methods, and variations such as backfilling (*Hussain et al., 2023*). These methods are computationally lightweight, making them attractive for real-time scenarios and systems where scheduling decisions must be made rapidly. However, heuristics tend to focus on local optimization and often ignore global workload distribution. This results in inefficiencies such as load imbalance, violation of QoS constraints, and poor scalability when workflows are large or highly interdependent.

To overcome the limitations of heuristics, numerous metaheuristic algorithms have been applied to workflow scheduling. Genetic Algorithms (GA) (*Kumar & Karthikeyan, 2024*), Particle Swarm Optimization (PSO) (*Fu et al., 2023*), Ant Colony Optimization (ACO) (*He et al., 2022*), and other evolutionary strategies (*Khiat, Haddadi & Bahnes, 2024*; *Tekawade & Banerjee, 2023*) are among the most widely used. These methods explore large solution spaces effectively and can optimize multiple objectives such as makespan, energy consumption, and cost (*Jingwei et al., 2023*). However, they are computationally expensive, suffer from slow convergence in large search spaces, and are highly sensitive to parameter tuning. These drawbacks make them less suitable for on-demand and real-time cloud environments where scheduling must be both accurate and fast.

More recently, machine learning (ML) and probabilistic approaches have been introduced to enhance adaptability in dynamic and uncertain environments. For example, reinforcement learning and deep learning models (*Mishra & Majhi, 2021*; *Shen et al., 2025*; *Zhou et al., 2022*) can predict task execution times, resource consumption, and make adaptive scheduling decisions. Similarly, probabilistic models (*Russo et al., 2024*; *Ye et al., 2022*) attempt to manage uncertainty by modeling task execution times and costs as distributions rather than fixed values. These approaches improve adaptability (*Singh, 2022*) and predictive accuracy, but they introduce high training costs, increased scheduling delays, and difficulties in guaranteeing deterministic compliance with strict user-defined deadlines or budgets (*Naqin et al., 2020*). As a result, their applicability is often limited to offline or batch scheduling scenarios rather than real-time cloud service provisioning.

Hybrid frameworks attempt to combine the strengths of multiple paradigms (*Sita, Bhambri & Kataria, 2023*). Examples include multi-objective optimization frameworks (*Hegde et al., 2024*; *Mohammadzadeh & Masdari, 2021*) that jointly consider makespan, energy consumption, and execution costs, as well as deadline and cost constrained evolutionary algorithms (*Tekawade & Banerjee, 2023*). While these frameworks achieve

better trade-offs among competing objectives, they often struggle when multiple QoS constraints must be satisfied simultaneously. Many assume simplified workflow structures (*Ara et al., 2020*), single global deadlines, or independent tasks, which reduce their applicability to real-world scientific workflows that are large, dynamic, and constraint rich.

The Hungarian Algorithm (HA) has attracted attention as an efficient method for solving assignment problems in polynomial time (*Alam et al., 2022*; *Juliet & Brindha, 2023*; *Lee, 2022*). Cloud scheduling (*Khiat, Haddadi & Bahnes, 2024*) has been applied to optimize resource allocation and minimize costs. HA-based methods are particularly appealing because they guarantee optimal task-to-resource assignments without the overhead of metaheuristic searches. However, most existing HA applications in cloud scheduling focus on pure cost or resource utilization optimization and fail to incorporate user-defined constraints such as deadlines and budgets. This significantly limits their use in real-world cloud workflow scheduling scenarios, where multi-constraint optimization is essential.

Several critical research gaps can be identified, many methods optimize one metric (*e.g.*, makespan, energy, or cost) at the expense of others, leading to QoS violations under realistic workloads. Metaheuristic and ML-based approaches often incur prohibitive overheads, limiting their practicality in deadline-sensitive and on-demand contexts. Existing Hungarian Algorithm applications do not adequately incorporate user-defined constraints such as deadlines and cost budgets. Few methods combine lightweight heuristic or evolutionary mechanisms with exact assignment algorithms (*Vásconez et al., 2024*), a combination that is critical for achieving scalability (*Ahmed, Choudhary & Al-Dayel, 2024*) while satisfying constraints.

Furthermore, numerous unresolved challenges remain in terms of security. Existing cloud environments remain vulnerable to cyberattacks, as client data stored within vendor-controlled infrastructures may not fully comply with security standards, exposing users to breaches and malicious assaults (*Elsayed, Almustafa & Gebali, 2022*; *Singh, Jeong & Park, 2016*). Moreover, classical cryptographic techniques are insufficient to resist emerging quantum attacks, which creates an urgent need for post-quantum cryptography (PQC) solutions to ensure future-proof protection of cloud systems (*Ukwuoma et al., 2022*). Although PQC is rapidly evolving, its integration into real-time workflow scheduling and resource allocation strategies remains underexplored (*Rajkumar et al., 2024*).

In addition, machine learning-based cloud management systems face a growing threat from poisoning attacks, including domain name system (DNS) cache poisoning and data manipulation. These adversarial strategies can compromise the decision-making process of scheduling and resource allocation by injecting malicious inputs into training datasets (*Mangalampalli & Karri, 2023*). While solutions have been proposed for anomaly detection (*Yang et al., 2024*) and lightweight cryptographic fault tolerance in advanced encryption standard (AES) (*Marisargunam, 2024*), they primarily address data integrity and cryptographic resilience (*Varnita et al., 2024*) but do not fully consider the multi-constraint optimization problem of workflow scheduling under strict deadlines and cost requirements.

To address these research gaps and challenges, this paper makes the following contributions:

- Proposes an integrated cloud task scheduling and allocating approach (FTPA-HA) that satisfies the user's deadline and budget constraints.
- Bridges performance and security perspectives: Unlike prior studies that address security or scheduling in isolation, our approach complements ongoing advancements in post-quantum cryptography and anomaly detection by introducing a scalable and constraint-aware workflow scheduling strategy that enhances resilience against malicious manipulations and resource misallocations.
- Improves scalability and adaptability: Through genetic-based crossover and mutation operators in FTPA, the proposed system explores a wider solution space while ensuring near-optimal allocation efficiency, outperforming state-of-the-art scheduling techniques across diverse workflow sizes and varying user constraints.

While prior research has identified the security vulnerabilities and computational inefficiencies of cloud systems, this study fills the gap by presenting a constraint-driven, security-conscious, and computationally efficient scheduling mechanism that ensures deadline-and budget-compliant task allocation in heterogeneous cloud environments.

## METHODOLOGY AND MATERIALS

Allocating cloud tasks to heterogeneous VMs can result in varying performance outcomes. Therefore, many different constraints are introduced in workflow scheduling computations to generate near optimal solutions, such as deadlines, cost, load balancing, and energy efficiency. However, the constraints that are considered in workflow scheduling depend on the nature or size of the job, VMs availability, and running environment. In this paper, we define the workflow scheduling boundaries that affect the deadline and cost constraints as follows:

- Makespan: the total time required to execute the entire workflow task. It represents the time duration of the application task completion.
- Financial cost: the value which incurred while running a workflow. The financial cost is defined as follows: Let $ET_{t_i}^{VM_q}$ represents task $t_i$ execution time on the virtual machine $VM_q$, $TT_{ij}$ represents the time of transfer data between task $t_i$ and task $t_j$, $cost_{vm_q}$ represents the cost of task $t_i$ running on a $VM_q$, and $cost_{transfer_{ij}}$ represents the transfer cost between the parent task $t_i$ and the child task $t_j$; then the total cost $Cost_W^{VM_{set}}$ of the workflow schedule is computed in Eq. (1)

$$Cost_W^{VM_{set}} = \sum_{i=1}^{n} \left[ \left( cost_{vm_q} \times \left( ET_{t_i}^{VM_q} \right) \right) + \left( cost_{transfer_{ij}} \times TT_{ij} \right) \right]. \quad (1)$$

- $VM$ lease time $LT(VM_p)$: the time when the virtual machine $VM_p$ becomes idle.
- Start time: the time at which task $t_i$ starts execution on its assigned virtual machine $VM_q$. Let $t_{root}$ represents the root, *i.e.*, entry task, Parents($t_i$) represents the parents of task $t_i$,

$ParentsFinishTime(t_i)$ represents the parents finish time task $t_i$; then the start time $ST_{t_i}^{VM_q}$ is computed in Eq. (2)

$$ST_{t_i}^{VM_q} = \begin{cases} 0, & if \ t_i = t_{root}, & otherwise \\ \max_{t_v \in Parent(t_i)} & (LR(VM_j), & \max(ParentsFinishTime(t_i) + TT_{ij})). \end{cases} \quad (2)$$

- Execution time: the time required for a task to be executed on the assigned virtual machine, which includes the running time and data transmission times. Let $Size(t_i)$ represents the size of task $t_i$ in millions of instructions, $Speed(VM_p)$ represents the processing power of the virtual machine $VM_p$ in millions of instructions per second MIPS, and $VM_{(p-PE)}$ represents the number of virtual machines $VM_p$ processing elements or cores; then the execution time is computed in Eq. (3)

$$ET_{t_i}^{VM_p} = \frac{Size(t_i)}{Speed(VM_j) * VM_{(p\_PE)}}. \quad (3)$$

- Finish time: the time at which task $t_i$ ends its execution on the assigned virtual machine. The finish time equals to the sum of task start time and its execution time.
- Depth: the topological level of the task $t_i$ in the workflow. The depth of task $t_i$ $Depth_{ti}$ is computed in Eq. (4).

$$Depth_{ti} = \begin{cases} 0, & if \ t_i = t_{root} \\ 1 + \max_{t_v \ \in \ predecessor(t_i)} & Depth(t_v) + 1 \end{cases} \cdot \quad (4)$$

- Deadline: represents the user defined deadline and calculated in Eq. (5)

$$Deadline = Time_{HEFT}^{W} \times (1 + \alpha), \quad (5)$$

where $Time^{W}{}_{HEFT}$ is the scheduling time of the tasks of the workflow W that is computed using the HEFT scheduling algorithm (*Gobichettipalayam, Sandhiya & Sruthi, 2023*). The parameter α is a randomly generated number that controls the deadline constraint. The task scheduling output must satisfy the constraint: $Schedule_{ti} \leq$ user defined Deadline.

- Budget: denotes the user defined budget and computed in Eq. (6)

$$Budget = Cost_{cheapest}^{W} \times (1 + \omega), \quad (6)$$

where $Cost^{W}{}_{cheapest}$ is the cost of scheduling the tasks of the workflow $W$, on the cheapest VM. The parameter ω is a randomly generated number that controls the budget constraint. The task scheduling output must satisfy the constraint: $Schedule_{cost} \leq$ User defined budget.

The objective of our proposed FTPA-HA scheduling approach is to minimize the makespan, which refers to the total execution time required to complete all workflow tasks on the allocated virtual machines. The makespan is primarily affected by the following key decision variables, task–VM assignment, task start times, and task execution durations.

The task–VM assignment variable ($x_{i,q}$) determines which virtual machine executes each task. Selecting faster or less-loaded VMs reduces execution time and, consequently, the overall makespan. The start time variable ($ST_i$) specifies when each task begins execution, constrained by the completion of its parent tasks and the associated data transfer times. The execution time variable ($ET_{i,q}$) depends on both the task size and VM computational capacity, while the finish time $FT_i$ defines each task's completion, with the largest finish time across all tasks determining the makespan.

Several constraints further influence this measure. The precedence constraint defined in Eq. (2) enforces task dependencies, ensuring that no task can start before its predecessors complete, directly shaping task scheduling order and duration. The execution time constraint defined in Eq. (3) links task size to VM performance, meaning that allocating tasks to slower VMs extends the makespan. The deadline constraint defined in Eq. (5) restricts the total execution time to remain within a defined limit, while the budget constraint defined in Eq. (6) caps the total cost, which may restrict the use of high-performance VMs and thus indirectly increase makespan. Additionally, the task depth constraint defined in Eq. (4) reflects workflow hierarchy; deeper dependency levels naturally prolong the overall completion time.

Generally, the makespan is determined by how effectively the proposed model balances task allocation, dependency handling, and resource utilization within the imposed cost and deadline limitations. Algorithm 1 describes the scheduling and allocating processes of the proposed approach.

In this proposed scheduling and allocating algorithm, the three-point crossover (*Singh, 2022*) genetic operator is used to create a new near-optimal task population solution that expands the task allocating search space. The mutation genetic operator (*Ahmed, Choudhary & Al-Dayel, 2024*) supports the allocation process with random probability that gives the tasks a low fitness value and the chance to produce new feasible solutions. This demonstrates the ability of the proposed approach to enhance task scheduling performance in cloud computing systems.

The proposed approach consists of two-stage scheduling algorithms, FTPA and HA. First, FTPA generates the most feasible set of workflow tasks under strict constraints, mitigating weaknesses of heuristic-only methods. Then, HA is applied to optimally assign these tasks to the fittest VMs in polynomial time. The reflection of both FTPA and HA on Algorithm 1 is described as follows:

The FTPA primary role is to generate the fittest workflow task population set under user-defined deadlines and cost constraints. Reaching the fittest task allocation in (Step 7) refers to the evaluation stage where the newly generated task population is assessed against the user-defined deadline and cost constraints. At this point, the algorithm determines whether the evolved set of candidate solutions includes a feasible population that satisfies both constraints while maintaining high fitness. If such population is not found, the algorithm returns to (Steps 2–6), continuing the crossover and mutation process until a suitable solution is produced. This implies that (Step 7) serves as a checkpoint ensuring that only constraint-compliant and near-optimal task populations progress to the assignment stage. This iterative process (Steps 1–7), ensures that infeasible or low-fitness

---

**Algorithm 1 Scheduling and allocating workflow tasks in cloud computing.**

Input: Cloud workflow tasks, user deadline constraint, user cost constraint
Output: The fittest tasks population set
Begin
    Step 1: Input the initial set of tasks population.
    Step 2: Select a task from the initial set of tasks population.
    Step 3: Perform the three-point crossover genetic operator on the selected task.
    Step 4: Apply the mutation genetic operator on the generated three-point crossover task.
    Step 5: Add the newly generated tasks to the new feasible task population.
    Step 6: Replace the initial tasks population with the new task feasible population.
    Step 7: Evaluate the new feasible population. If the fittest task allocation is not met, repeat steps 2–7.
    Step 8: If the fittest task allocation is met, add it to the fittest tasks population ($n \times n$) matrix and determine the minimum element in each row and deduce it from each element in that row.
    Step 9: Determine the minimum element in each column and deduct it from each element in that column.
    Step 10: Determine the minimum number of lines to cover all zero elements in the matrix.
    Step 11: Determine the smallest element (k) that is not covered by a line. For each element that is covered twice, add k and subtract k from the elements that are left uncovered.
    Step 12: Allocate the smallest element (k) to the fittest Virtual Machine.
End

---

allocations are eliminated while retaining solutions that balance execution time, financial cost, and deadline adherence.

The second stage of the model applies the HA to optimally assign the refined FTPA-generated tasks to the fittest available virtual machines (VMs). HA operates on the cost matrix by systematically reducing rows and columns, covering zeros with minimal lines, and adjusting uncovered elements until the optimal assignment is achieved in polynomial time. This ensures that each task is allocated to the VM that minimizes total execution cost while still meeting deadline constraints. The HA is integrated explicitly in (Steps 8–12) as the second stage of the proposed scheduling algorithm, where the mapping of tasks to VMs are completed. These steps detail the core phases of the HA: constructing the ($n \times n$) cost matrix from the fittest task population (Step 8), row and column reductions to normalize costs (Step 9), covering all zeros with the minimum number of lines (Step 10), adjusting the uncovered elements to generate additional zeros (Step 11), and finally assigning tasks to the fittest virtual machines by selecting the optimal zero-cost entries (Step 12). Together, the two stages form a complementary process: FTPA filters and prepares the most promising candidate task sets, while HA guarantees cost-effective and deadline-compliant allocations demonstrating the novelty and effectiveness of the proposed scheduling approach.

## EXPERIMENTAL RESULTS AND PERFORMANCE ANALYSIS

The proposed approach is simulated in Azure Application Service to deploy the assignment problem using Java 17 with Visual Studio on a laptop machine (Core i7 CPU, 2.40 GHz, and 8 Giga Byte (GB) RAM), and a set of different VMs running on heterogeneous cloud resources. The data set represents the workflow tasks costs and consists of 20 ($n \times n$) matrices. The matrix rows represent agents (*e.g.*, workers, machines, *etc.*) and the columns represent tasks (*e.g.*, jobs, resources, *etc.*). Each matrix cell (rowi,

colj) value represents the cost of allocating task (j) to the agent (i). The tasks allocating costs are used by the Hungarian Algorithm (the second stage of the proposed approach FTPA-HA) to find the fittest VM that best allocate each task. To assess how well the HA assigns tasks to VMs, correctly and optimally, we define the workflow allocating boundaries that affect the performance of the workflow constraints, namely Precision and Recall that are measured in terms of the following metrics:

- True positive (TP): The number of tasks to which the best VM is successfully allocated.
- False positive (FP): The number of not correct allocations assigned to VM.
- False negative (FN): the number of missed allocations.

  Precision and Recall are defined as follows:

- Precision is the percentage of tasks that are correctly predicted positive allocations to the total predicted allocations and computed in Eq. (7):

$$Recall = TP \div (TP + FN) \tag{7}$$

- Recall is the percentage of tasks that are correctly predicted positive allocations to all actual allocations and computed in Eq. (8):

$$Precision = TP \div (FP + TP). \tag{8}$$

# DISCUSSION

The proposed scheduling approach is validated by conducting several experiments that carried out in the same configuration and execution environment in order to guarantee a fair and reasonable comparison with well-known scheduling algorithms such as Optimal Sequence Dynamic Assignment Algorithm (OSDAA) (*Kumar, Surachita & Kumar, 2022*), Round-Robin (R-R) (*Balharith & Alhaidari, 2019*), Random (RD) (*Manikandan, Gobalakrishnan & Pradeep, 2022*), Genetic Algorithm (GA) (*Kumar & Karthikeyan, 2024*), and Particle Swarm Optimization (PSO) (*Fu et al., 2023*). The main objective of all approaches in comparison was to reduce the makespan. Simulations were performed using a standardized testbed equipped with an Intel Core i7 processor (2.4 GHz, 8 cores), 8 GB RAM, and running Ubuntu 20.04. Experiments deployed 10 virtual machines (VMs), each configured with 20 vCPUs, 21.4 GB RAM, 13,000 MHz processor speed, 100 GB storage, and 0.000001 transfer cost unit/s. The workload consisted of 200 tasks, either synthetically generated with directed acyclic graph (DAG) dependencies or drawn from standard scientific workflows, with task sizes (1,000–5,000) million instructions. All scheduling methods were tested under non-preemptive execution with no dynamic task arrivals, ensuring consistency across all evaluations. While the OSDAA configuration followed the default parameters, each of GA, PSO, R-R, and RD scheduling approaches need specific settings and tuning for best optimization. The GA was configured with a population size of 50, crossover rate of 0.8, mutation rate of 0.1, and 100 generations. The PSO used a swarm

size of 30 with inertia weight 0.7, cognitive weight 1.5. The R-R and RD approaches were assigning tasks randomly without priorities.

The reinforcement learning-based strategy is used in the second stage (HA allocation) of the proposed approach to improve adaptability. After each scheduling decision, the system receives feedback on allocation performance (*e.g.*, whether deadlines and cost constraints were met). This feedback is then used to update the allocation policy, enabling the algorithm to learn from prior scheduling outcomes. Over time, this reinforcement process enhances the Hungarian algorithm's effectiveness by guiding it toward more efficient task-to-VM assignments under dynamic cloud conditions.

The heuristic-based strategy is realized in the FTPA stage of the proposed approach. In this stage, heuristic-inspired genetic operators, specifically three-point crossover and mutation, are employed to generate feasible task populations that satisfy user-defined deadlines and cost constraints. This heuristic process enables the scheduler to find near-optimal solutions efficiently without exhaustively searching the entire solution space. In addition, the heuristic-based strategy provides computational efficiency, ensures scalability in large-scale on-demand cloud service provisioning, and offers high-quality candidate solutions for the Hungarian Algorithm to refine.

The simulation parameters and the VMs configurations are summarized in Tables 1 and 2. Better cloud computing efficiency is achieved by having the scheduler distribute jobs to earlier VMs based on the task's information and the resource information server with minimal makespan value.

Cloud providers offer flexible memory allocation, so memory in the cloud is elastic and scalable, allowing resources to scale up or down in response to demand, preventing memory waste or shortages for applications. The makespan fitness function is computed in Eq. (9):

$$Makespan^{\min} = max\{ET_{t1}, ET_{t2}, ET_{t3}, \ldots, ET_{tn}\} \tag{9}$$

where $ET$ is the execution time needed to execute ($tn$) task on VM. Makespan reflects the overall efficiency of the scheduling workflows of cloud computing approaches. In the proposed scheduling FTPA-HA approach, the overall competence goal is minimizing the makespan to optimize resource use and task throughput under user constraints, deadline and cost.

The key factors that influence makespan in cloud-based scheduling and considered in our approach are population size, fitness weight, robust weight, crossover rate, mutation rate, input matrix size, input matrix cost. Table 3 summarizes the range and methods of the key factors changes and their impact on the makespan.

The proposed scheduling approach FTPA-HA effectively reduces makespan by combining the FTPA (stage 1) that generates decent initial solutions respecting cloud constraints, such as deadline, cost, and virtual machines availability, and HA (stage 2) that refines the solution for balanced parallel execution, minimizing idle virtual machines time and bottlenecks. This two-stage scheduling approach significantly reduces makespan, improving resource utilization and quality of service QoS, thus, ensure both efficient and

**Table 1 FTPA simulation parameters.**

| Workflow type | Scientific Direct Acyclic graph (DAG) workflow |
|---|---|
| Workflow name | Montage, Epigenomics, Ligo-Inspiral, Sipht, and CyberShake |
| DAG size | 100, 200, 300, 400, 500, 600, 700, 800, 900, 1,000 tasks |
| Number of edges | 456, 4511 |
| Average data tasks size | 2.7 MB, 3.1 MB, 3.5 MB, 3.7 MB |
| Resource pool | 10 spaceShared VMs Data Center |
| Number of VMs | 10 |
| Transfer cost unit/s | 0.000001 |
| Max. generation | 300 |
| Population size | 200 |
| Crossover probability | 50% |
| Mutation probability | 10% |

**Table 2 VMs configuration parameters.**

| VM type | VM cores | Processor speed (MHz) | Memory (GB) | Cost unit/s |
|---|---|---|---|---|
| $VM_{type}1$ | 1 | 2,000 | 1.8 | 0.000012 |
| $VM_{type}2$ | 4 | 4,500 | 5.3 | 0.000036 |
| $VM_{type}3$ | 5 | 8,000 | 8 | 0.000049 |
| $VM_{type}4$ | 8 | 10,000 | 16 | 0.00010 |
| $VM_{type}5$ | 20 | 13,000 | 21.4 | 0.00014 |

practical scheduling in dynamic cloud environments. Table 4 shows the makespan results of the scheduling methods.

The results depicted in Table 4 show that the proposed two-stage FTPA-HA approach significantly outperforms existing approaches in terms of computational time since the first stage of the proposed approach is a computationally efficient lightweight heuristic-based scheduling technique used to optimize resource allocation that may not always provide optimal solutions but aims to find fast and good enough near optimal solutions, making it ideal for large-scale and on-demand cloud service provisioning with resource constraints where decisions should be made in near real-time. The second stage where the HA is utilized to allocate workflow tasks to the fittest VM in polynomial time. This is a reinforcement learning-based scheduling technique that supports feedback after each scheduling decision and updates its plan to improve its future allocations in polynomial time, making it more efficient for learning complex environments which can adapt to changing tasks and constraints over time, thus resulting in better long-term optimization. In addition, we carried out the experiments under the following execution constraints: no task anticipation, no dynamic task arrival, and fixed deadline which is not considered in makespan evaluation.

To evaluate the efficiency of state-of-the-art cloud workflow scheduling approaches, a comparative analysis was conducted between the proposed FTPA-HA scheduling approach and two state-of-the-art scheduling algorithms namely, CEDCES: A

**Table 3 The impact of range and methods of key factors on the makespan.**

| Parameter | Range | Methods of change | Mechanism of impact on makespan |
|---|---|---|---|
| Population size | 100–1,000 | Linear increments | Larger populations → more balanced loads and shorter makespan. |
| Fitness weight | 0.001–0.1 | Multi-objective weighting | Emphasis on load balancing *vs.* sensitivity shifts optimization bias, shaping the scheduling output that flows into our approach—stage 2. |
| Robust weight | 0.1–10 | Log-scale adjustments | Heavily penalizing uncertainty → increases resource under utilization→ increase makespan. |
| Crossover rate | 0.6–0.95 | Step changes | Too high → enhances recombination of decent traits but can reduce privileged solutions → affects convergence speed and diversity. |
| Mutation rate | 0.01–0.2 | Step changes | Too low → early convergence. Too high noise. |
| Input Size ($n \times n$ matrix) | Derived from the proposed FTPA—stage 1 | Indirect *via* the proposed FTPA—stage 1 outputs | Suboptimal assignments → extended makespan. |
| Input cost matrix precision | Small–large granularity | Normalized scaling | Minimizes total execution time → reduce makespan. |

**Table 4 Cloud workflow scheduling methods comparison in terms of makespan.**

| Scheduling method | Makespan (s) | Number of scheduling tasks |
|---|---|---|
| Proposed method | 0.00014 s | 200 |
| OSDAA | 11,000 s | 200 |
| Round-Robin (R-R) | 17,000 | 200 |
| Random (RD) | 17,000 | 200 |
| GA | 10,000 | 200 |
| PSO | 20,000 | 200 |

Cost-Effective Deadline Constrained Evolutionary Scheduler for Task Graphs in Multi-Cloud System (*Mangalampalli & Karri, 2023*) and GATES: Cost-aware Dynamic Workflow Scheduling *via* Graph Attention Networks and Evolution Strategy (*Shen et al., 2025*). This study implements the three scheduling approaches on five real-world scientific applications, Cybershake (*Tang et al., 2021*), Montage (*Singh, Jeong & Park, 2016*), Ligo-Inspiral (*Elsayed, Almustafa & Gebali, 2022*), Epigenomics (*Sadhasivam, Balamurugan & Pandi, 2018*) and Sipht (*Youseff, Butrico & Silva, 2008*). Each scientific application is tested on workflow task sizes (100, 200, …, 1,000) with different deadlines and costs constraints. Figures 1 and 2 present an average of 10 experimental results of the proposed FTPA-HA performance efficiency against CEDCES and GATES approaches in terms precision and recall with deadline and cost constraints.

Figure 1 shows that the CEDCES precision values under deadline constraint drop from 92.0% with 100 tasks to 87.3% with 1,000 tasks, and under cost constraint drop from 93.0% with 100 tasks to 86.6. This figure shows also that the GATES precision values under deadline constraint drop from 94.0% with 100 tasks to 89.3% with 1,000 tasks, and under cost constraint drops from 93.1% with 100 tasks to 88.7% with 1,000 tasks.

In addition, this figure shows that the proposed FTPA-HA precision values under deadline constraint drop from 97.2% with 100 tasks to 92.8% with 1,000 tasks, and under

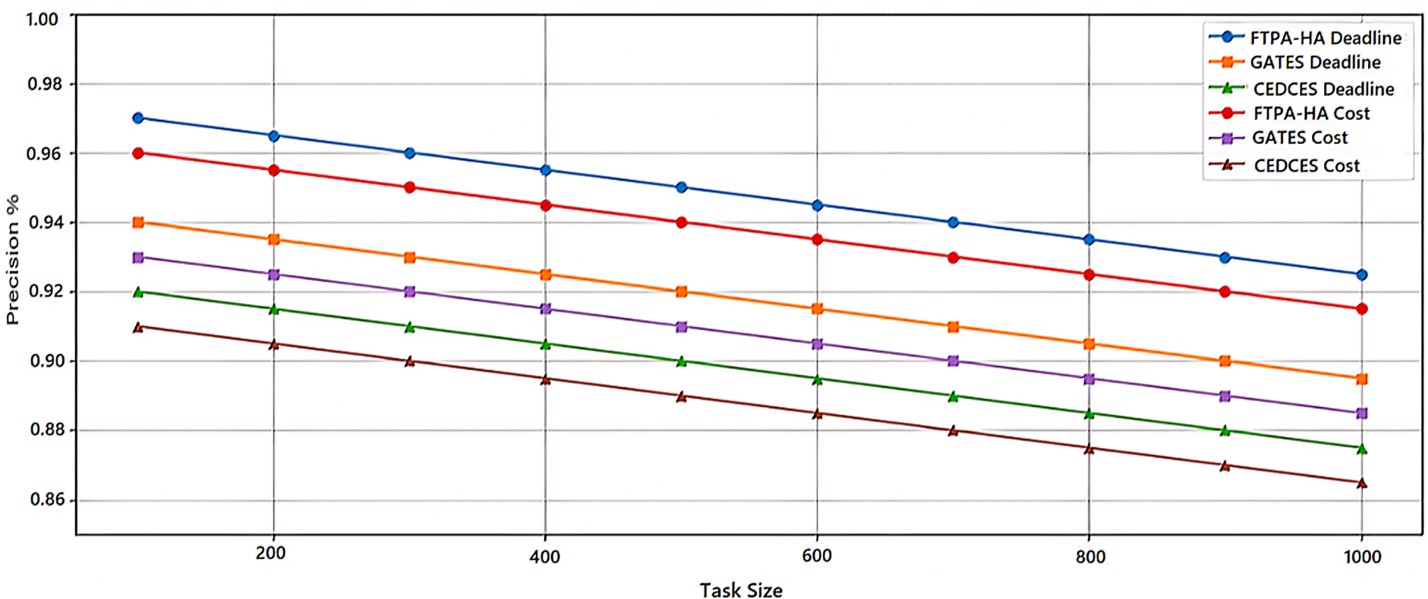

**Figure 1 Precision performance of the scheduling approaches.**

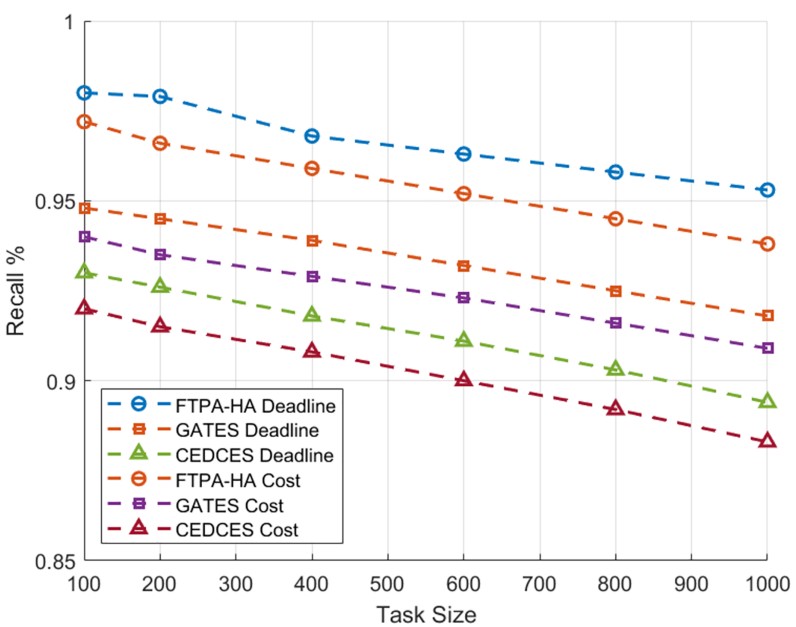

**Figure 2 Recall performance of the scheduling approaches.**

cost constraint drops from 96.0% with 100 tasks to 91.7% with 1,000 tasks. Figure 2 indicates that the CEDCES recall values under deadline constraint drop from 93.0% with 100 tasks to 88.7% with 1,000 tasks, and under cost constraint drop from 92.0% with 100 tasks to 87.6% with 1,000 tasks. Besides, this figure shows that the GATES precision values under deadline constraint drop from 95.1% with 100 tasks to 90.8% with 1,000 tasks, and

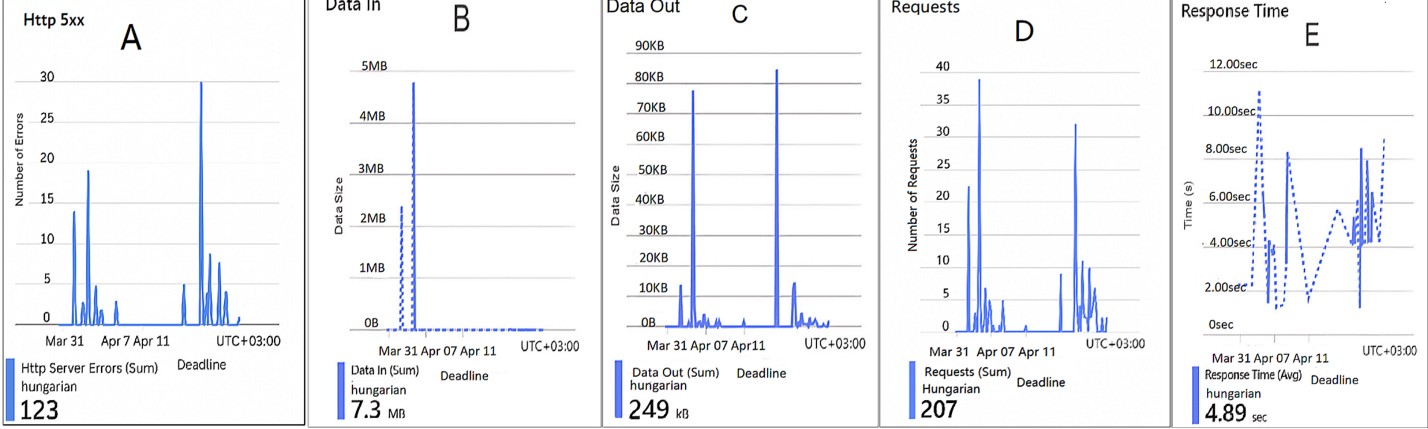

**Figure 3 Results of the assignment problem for the first configuration.** (A) The number of HTTP 5xx server-side errors. (B) The total amount of data received by the server. (C) The total amount of data sent from the server. (D) The total number of HTTP requests received. (E) The average server response time in seconds.

under cost constraint drops from 94.0% with 100 tasks to 89.4% with 1,000 tasks. Furthermore, this figure shows that the proposed FTPA-HA precision values under deadline constraint drop from 98.0% with 100 tasks to 93.5% with 1,000 tasks, and under cost constraint drops from 97.2% with 100 tasks to 92.7% with 1,000 tasks.

The experimental results confirm that CEDCES approach struggles to manage increasing task complexity, especially under tight deadline and cost constraints, and the GATES technique improves adaptability but lacks optimal refinement under multi-objective constraints deadline and cost. On contrast, the proposed approach FTPA-HA distinguished by task-resource allocating that leads to optimal resource utilization (generates high precision values) and lower failure rates (generates high recall values) across all task sizes. However, the FTPA-HA performance depends on well-tuned mutation and three crossover rate parameters that degrade the performance slightly under increasing constraints but prove its adaptability even workloads scale high. This makes the proposed FTPA-HA a robust approach for large-scale, on-demand cloud task scheduling in deadline-sensitive and cost-limited situations.

To illustrate the second stage of the proposed approach, we implement the Hungarian algorithm on a web application hosted in a cloud cluster named Hungarian and record the system activity and performance in the period March 31–April 7, 2025.

Figures 3, 4, 5, 6, 7, and 8 represent two sets of performance monitoring data for two different users with different configurations. In the first dataset, the HA is implemented with various performance configuration metrics: Hypertext Transfer Protocol (HTTP) 5xx server-side errors, total amount of data received by the server, total amount of data sent from the server, number of HTTP requests received, and the average server response time in seconds. In the second dataset, the HA is utilized with different performance configuration metrics. This comparison exposes differences in traffic intensity, reliability, and response efficiency. The datasets features are described as follows: HTTP 5xx errors represents server-side HTTP error codes (500–599), indicating issues with the server when

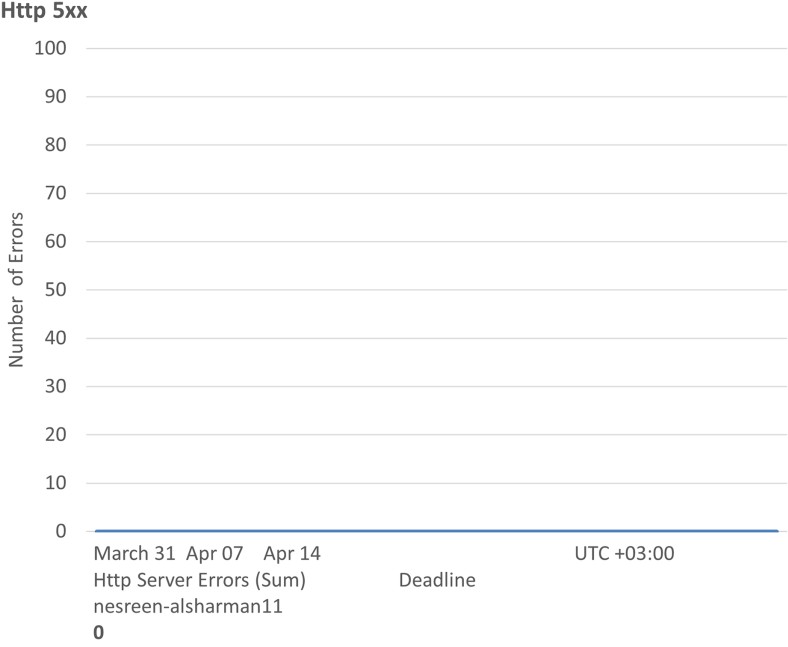

**Figure 4  Results of the Http 5xx of the second configuration.**

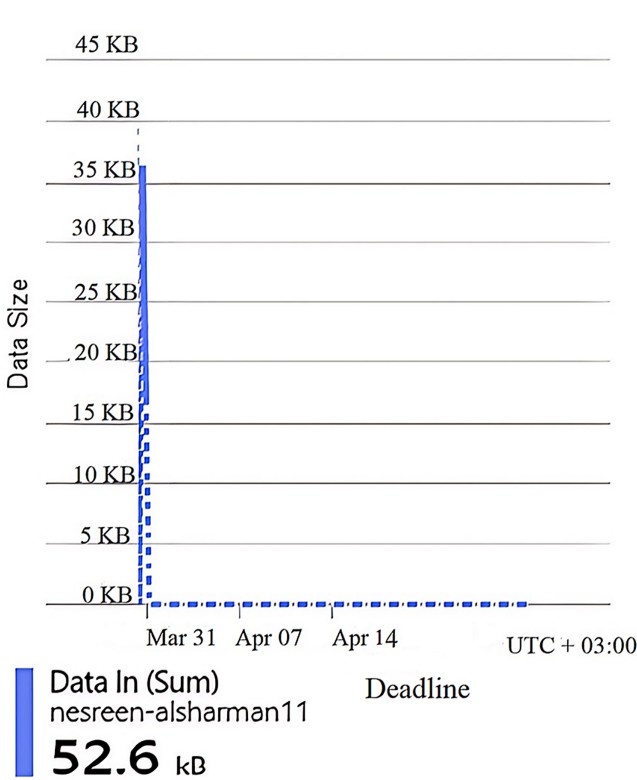

**Figure 5  Results of the data in of the second configuration.**

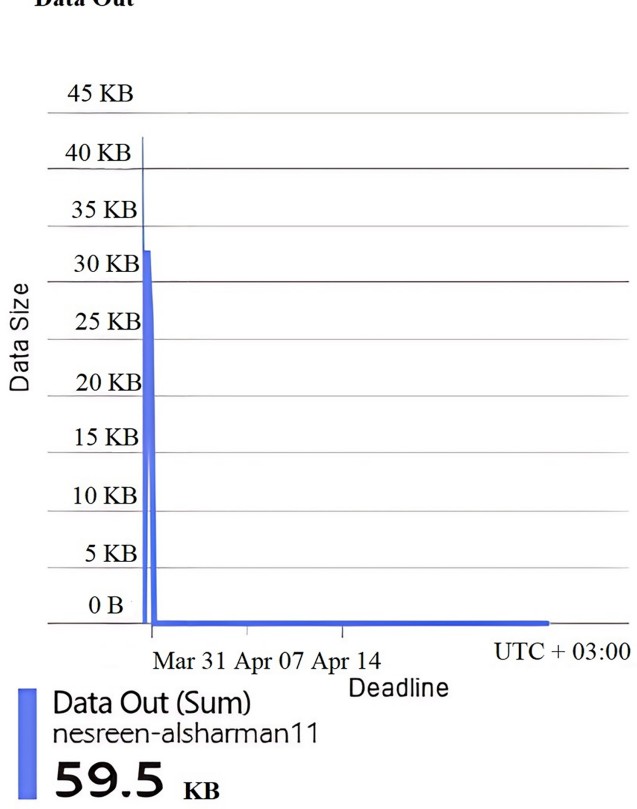

**Data Out**

**Figure 6 Results of the data out of the second configuration.**

processing requests. Data In denotes the amount of incoming data received by the server from users. Data Out was defined as the amount of outgoing data sent by the server to users. Requests represent the number of requests made to the server. Response Time expresses the average time taken by the server to respond to a request.

Figure 3 represents five performance metrics monitoring graphs related to a web application hosted in a cloud cluster named Hungarian. This figure provides experimental records of system activity and performance in the period March 31–April 7, 2025. The figure sub-graphs are described as follows: Figure 3A represents the number of HTTP 5xx server-side errors. There are several noticeable spikes in 5xx errors, peaking at around 30 errors at one point, and 123 errors are spread unevenly over the timeline, indicating intermittent server-side issues and backend server instability due to crashes, timeouts, or server overloads, potentially correlating with request spikes. Figure 3B represents the total amount of data (7.3 MB) received by the server. There is one significant spike, reaching around 5 MB at a single point and the rest of the timeline shows almost negligible data input. A single large spike implies a large request payload during that instance. Figure 3C represents the total amount of data (249 kB) sent from the server, mirroring the request pattern. There are multiple small spikes, maxing at around 80–90 kB. Little data is sent out by the server suggesting lightweight responses, such as text-based or small JavaScript

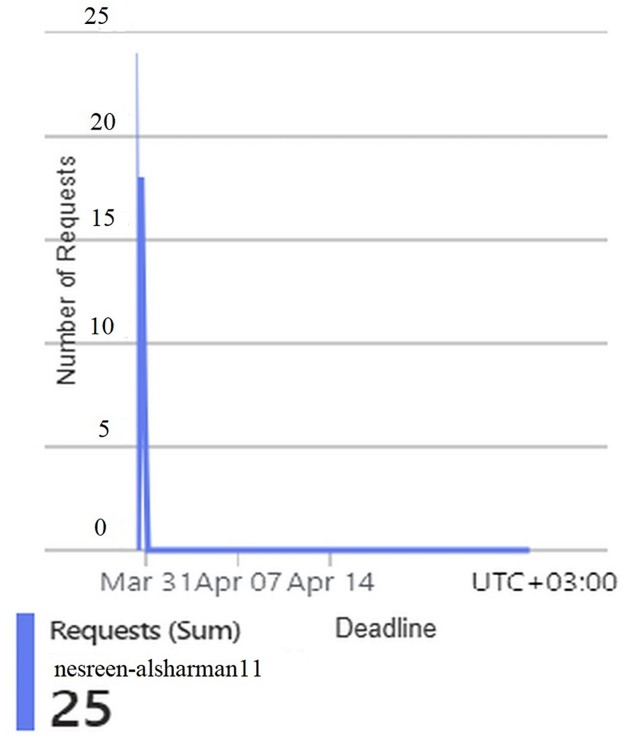

**Figure 7  Results of the requests of the second configuration.**

Object Notation (JSON) payloads. Figure 3D represents the total number of HTTP requests received (207). Several spikes in requests, with a peak of around 35–40 requests at once. High traffic contributes to error rates (similar pattern to 5xx errors). The request volume is moderate but shows intermittent behavior that might be causing performance degradation and triggering 5xx errors. Figure 3E represents the average server response time in seconds. Highly variable, with peaks nearing 12 s. A few drops below 2 s, but the average remains high (4.89 s) aligns with request spikes and server errors. It may imply slow database queries, backend bottlenecks, or overloaded infrastructure. All sub-graphs in this figure show that the Hungarian algorithm handles more traffic but struggles with server errors and performance degradation during the implementation period.

Figures 4, 5, 6, 7, and 8 represent another set of performance monitoring charts that show different behaviors for the same web application by changing the performance parameters' values. It spans the same period (March 31–April 7, 2025). The figure sub-graphs are described as follows: Figure 4 represents zero HTTP 5xx errors that implies the server-side code for this system is running smoothly, without any crashes during the running period. Figure 5 represents the total amount of data (52.6 kB) received by the server. There is one small spike around 50 kB, which likely indicates minimal data

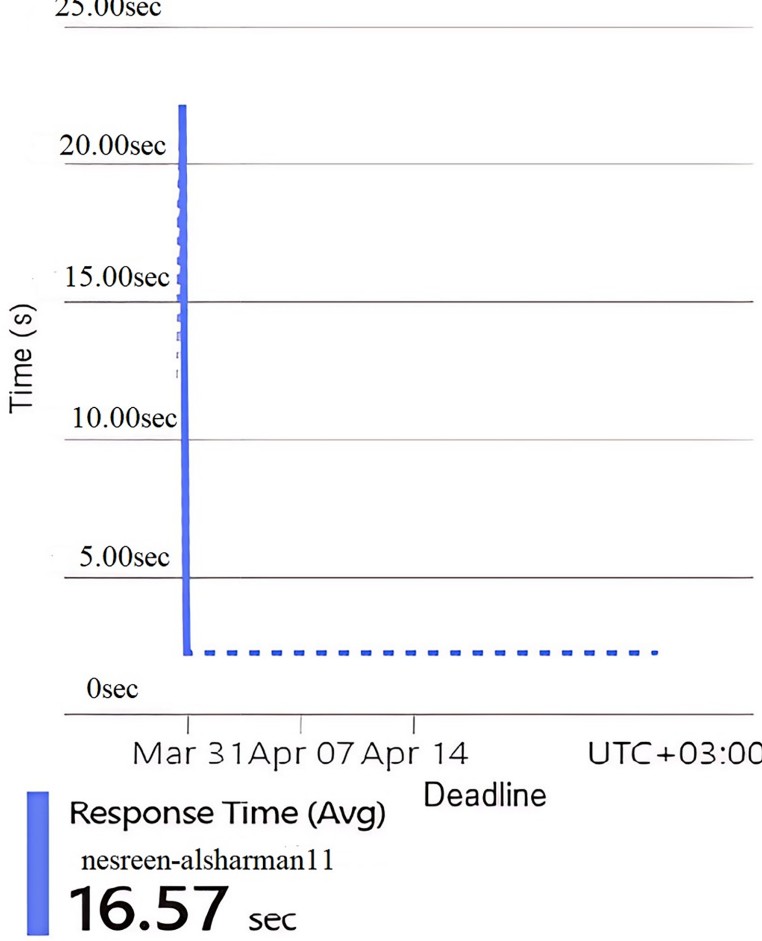

**Figure 8 Results of the response time of the second configuration.**

processing or low upload activities of the system. Figure 6 represents the total amount of data (59.5 kB) sent from the server. There is a little data out spike, reaching around 60 kB. This implies that the server is processing relatively small responses such as text-based or lightweight data. Figure 7 represents the HTTP requests received (25) during the implementation period. Low requests with a slight increase in traffic, indicating that this system received very few interactions.

This implies that the second configuration of the HA is not frequently used, or it might represent a testing environment with few interactions. Figure 8 represents the average server response time in seconds. The response time is extremely high at one point, peaking at 16.57 s. Despite this spike, the overall behavior seems reasonable and indicating no major variabilities follow this spike that indicate a slow request during the implementation period due to resource contention or a heavy operation. All parts in this figure show that

the Hungarian algorithm is underutilized but suffers from slow response times when it does receive requests.

## CONCLUSION AND FUTURE WORK

The Hungarian Algorithm works by determining the optimal assignment of tasks to VMs in such a way that the total cost is minimized while meeting deadlines and cost constraints. This was performed by creating a cost matrix that represented the cost of assigning each task to each VM and then finding the assignment that results in the lowest total cost. By incorporating the Fittest Task Population Algorithm into the process, the system can efficiently manage the population of tasks and ensure that only the most suitable tasks are considered for allocation. This helps in improving the overall performance of the cloud task scheduling environment by optimizing the allocation of tasks based on their specific requirements. The combination of the Hungarian Algorithm and the Fittest Task Population Algorithm allow for an efficient and erective task allocation process in cloud task scheduling environments, ultimately leading to better utilization of resources and improved quality of service for users. Furthermore, we conducted a sensitivity analysis to investigate the impact of different factors on the performance of the proposed approach. We varied parameters, such as the task arrival rate, resource availability, and network latency to determine the robustness and scalability of the proposed algorithm. The results show that our approach can adapt to changing conditions and effectively manage workload in dynamic and unpredictable environments. To further validate the effectiveness of our approach, we conducted case studies with actual cloud service providers and validated our algorithm in real-world scenarios. Results of these case studies confirmed the applicability and efficiency of our approach in improving cloud system performance and meeting user requirements. We aim to continue evolving and refining our scheduling approach to meet the growing demands and challenges of cloud computing environments, while providing cost-effective and reliable solutions for cloud customers. In future work, metaheuristic optimization algorithms such as Sea Lion Optimization (SLnO) (*Mell & Grance, 2011*) and Gray Wolf Optimizer (GWO) (*Zhang, Cheng & Boutaba, 2010*) will be investigated for more task scheduling constraints that enhance cloud service quality and match cloud balancing.

### Funding

This work was supported by German Jordanian University (GJU). The funders had no role in study design, data collection and analysis, decision to publish, or preparation of the manuscript.

### Grant Disclosures

The following grant information was disclosed by the authors:
German Jordanian University (GJU).

## Competing Interests

The authors declare that they have no competing interests.

## Author Contributions

- Nesreen Alsharman conceived and designed the experiments, performed the experiments, performed the computation work, prepared figures and/or tables, and approved the final draft.
- Ismail Hababeh conceived and designed the experiments, performed the experiments, performed the computation work, prepared figures and/or tables, and approved the final draft.
- Mohammad Alqudah analyzed the data, prepared figures and/or tables, authored or reviewed drafts of the article, and approved the final draft.
- Kholoud Nairokh analyzed the data, prepared figures and/or tables, authored or reviewed drafts of the article, and approved the final draft.
- Deefallah Alshorman analyzed the data, prepared figures and/or tables, authored or reviewed drafts of the article, and approved the final draft.

## Data Availability

The raw data are available in the Supplemental File.

## Supplemental Information

Supplemental information for this article can be found online at http://dx.doi.org/10.7717/peerj-cs.3385#supplemental-information.

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
