# Peer review of "Allocating constraint tasks schedules to promote on-demand cloud services by utilizing the Hungarian Algorithm"

_PeerJ Computer Science, doi:10.7717/peerj-cs.3385_

## Round 0.1 · original submission · Major Revisions

· Academic Editor

Major Revisions

**Language Note:** The review process has identified that the English language must be improved. PeerJ can provide language editing services - please contact us at [email protected] for pricing (be sure to provide your manuscript number and title). Alternatively, you should make your own arrangements to improve the language quality and provide details in your response letter. – PeerJ Staff

Reviewer 1 ·

Basic reporting

The overall framework is smooth, and the scientific significance is good.

Experimental design

The design has reached the par, and the performance can be demonstrated.

Validity of the findings

Some findings can be concluded in the current version.

Additional comments

The paper “Allocating constraint tasks schedules to promote on-demand cloud services by utilizing the Hungarian algorithm” presents a two-stage algorithm to handle multiple cloud workflow scheduling constraints. The Fittest Task Population Algorithm (FTPA) generates a cost-aware task population model that adheres to customer task cost constraints. This model serves as a foundation for the subsequent assignment of tasks to suitable virtual machines, thus optimizing cloud workflow scheduling. The Hungarian Algorithm, on the other hand, focuses on task allocation. It assigns each task from the population set generated by FTPA to the most appropriate cloud virtual machine (VM), aiming to minimize the overall task allocation cost. The experimental results demonstrate the effectiveness of the proposed approach. Overall, several issues can be considered to improve the quality.
1.It is recommended that the authors comprehensively review past literature to clarify the research status of similar two-stage methods. In the section on related work, the authors failed to investigate and expound on the existence of similar two-stage task scheduling methods.
2. The authors should supplement the detailed definitions of multiple constraints and clarify the constraint conditions in the form of formulas. For example, specific quantification methods and the interrelationships of constraints such as cost, time, and resource utilization should be defined.
3.It is recommended that the authors provide an overall flowchart of the article's algorithm to give readers a clearer understanding of the algorithm.
4. The authors should supplement the specific values of recall and precision under different experimental conditions and display their changing trends in the form of charts. Meanwhile, a detailed comparative analysis with the comparison algorithms is recommended to illustrate the advantages and limitations of the proposed algorithm under different constraint conditions.
5.It is recommended that the authors incorporate recently proposed and representative cutting-edge algorithms for comparative analysis.
6. In the "discussion" section, the authors mainly focused on the makespan and sensitivity analysis, but the data and technical support provided were insufficient. Regarding the makespan, only the numerical values of different algorithms were compared, and the key factors affecting the makespan and their relationships with the algorithm design were not deeply analyzed. In terms of sensitivity analysis, the range and methods of parameter changes, as well as the specific mechanisms of their impacts on the algorithm performance, should be explained in detail.
7.It is recommended that the authors carefully check the language expression of the article to ensure clear logic and well-organized content.

·

Basic reporting

1. In Tables 4 and 6, what does the multiplication sign indicate?

Experimental design

1. In Figures 1 and 2, what do the axes indicate?
2. There is no comparison with existing approaches.
3. There is no proper explanation for these figures.
4. Explain the datasets and their features.

Validity of the findings

1. Validate the proposed model through statistical analysis or convergence analysis.

Reviewer 3 ·

Basic reporting

1. The contribution of the work is unclear. If the proposed Algorithm 1 is where the novelty holds, authors are expected to explain what exactly is novel compared to the state of the art.
2. The repetition of bullet points for explaining the Hungarian Algorithm below Algorithm 2 should be revised or removed.
3. It is unclear how Algorithms 1 and 2 are integrated.

Experimental design

1. The Hungarian Algorithm is a well-known algorithm, and it is unnecessary to present the example using Tables 1-6 in Section 2.
2. It should be explained how the selected algorithms for comparison were configured, such as the objective function, prediction horizon, etc., especially when the results in Table 9 show a very drastic difference between the proposed method and the selected algorithms.

Validity of the findings

1. Table 9 shows that the proposed method achieved a makespan of 0.00014 seconds, approximately 1e-8 of the 10,000 seconds required by the GA algorithm, representing the smallest makespan among all the compared techniques. This result shows that the proposed method significantly outperforms existing solutions in terms of computational time. However, the configuration details of the experiment and the underlying reason for such a drastic improvement is not clearly explained.
2. Figures 1 and 2 need to be explained and guided to show the insights of the results.

---

## Round 0.2 · Major Revisions

· Academic Editor

Major Revisions

One of the reviewers raised a number of problems that the authors need to fix.

Reviewer 1 ·

Basic reporting

-

Experimental design

-

Validity of the findings

-

Additional comments

I think the revision is satisfactory.

Reviewer 3 ·

Basic reporting

1. The article’s expression is unclear, and numerous grammatical errors are present throughout. For example:
a. The defections above Equation (1) lack commas.
b. Equation (1): TT_{ij} is undefined.
c. “Equation1”
d. “equation 2”
e. Above Equation (4), the “i” in “ti” should be a subscript.
f. “The parameter is”
g. “Equation6”
h. Extra bullet points on above and below Equation (7).
i. “86.6This” on page 12

2. The literature review reads as a series of unrelated summaries, each describing a specific paper. It should instead be structured to help readers clearly understand the state of the art, the research gaps, and how this paper contributes to addressing those gaps.

Experimental design

1. The proposed method appears to be summarized in both Figure 1 and Algorithm 1, but both contain the same 12 steps. Such duplication is not appropriate for a technical paper. Additionally:
a. It is claimed that “The FTPA-HA consists of two algorithms, the FTPA that generates the fittest workflow tasks population set under users’ deadlines and cost constraints, and the HA, which is utilized to assign each task to the fittest VM.” However, it is unclear how either algorithm is reflected in Figure 1 or Algorithm 1.
b. The meaning of “Step 7: Fittest task allocation is met” is unclear, and no explanation can be found in the paper.
c. Steps 8-12 seem to be an implementation of the Hungarian algorithm. This should be explicitly presented to echo the aforementioned claim.

2. The definitions provided at the beginning of Section 1 do not appear to be used later in the paper.

3. The paper claims that the objective is to minimize the makespan, but this is not well elaborated what the decision variables and constraints are.

4. In Section 3:
a. In the first paragraph, it says “ The proposed approach uses the reinforcement learning-based strategy….”, but it is never explained how the reinforcement learning-based strategy is used or formulated.
b. The second paragraph says “A heuristic-based strategy is also used in the proposed approach….”. What exactly is this “heuristic-based strategy”?
c. The third paragraph only has one sentence.
d. The beginning of the fourth paragraph redefines “makespan”.
e. Extra spacing below the caption of Table 1.
f. The captions of Tables 3 and 4 should be on top of the tables.

Validity of the findings

1. Figure 3 is blurred, and the font size is too small to read. The caption of Figure 3 also ends with an extraneous quotation mark.

2. The paragraph below Figure 3 should be divided into two paragraphs, beginning a new paragraph with “To evaluate the efficiency of….”

3. Figure 5 is blurred, and the information presented is not well organized.

4. In the last paragraph on page 14, the phrase “This implies that the application is not…” is unclear because it is not specified what “the application” refers to.

5. The last paragraph of Section 3 reads like an opening paragraph and appears misplaced. It would be more appropriate at the beginning of the discussion for Figures 4 and 5.

---

## Round 0.3 · accepted · Accept

· Academic Editor

Accept

The authors correctly addressed the requested issues and therefore I can recommend this article for acceptance.

·

Basic reporting

Authors have addressed all my concerns.

Experimental design

Good

Validity of the findings

Impactful

Additional comments

Nil